# EDITING PERSONALITY FOR LLMS

## ABSTRACT

This paper introduces an innovative task focused on editing the personality traits of Large Language Models (LLMs). This task seeks to adjust the models' responses to opinion-related questions on specified topics since an individual's personality often manifests in the form of their expressed opinions, thereby showcasing different personality traits. Specifically, we construct a new benchmark dataset **PersonalityEdit** to address this task. Drawing on the theory in Social Psychology (Goldberg, 1990), we isolate three representative traits, namely NEUROTICISM, EXTRAVERSION, and AGREEABLENESS, as the foundation for our benchmark. We then gather data using GPT-4, generating responses that not only align with a specified topic but also embody the targeted personality trait. We conduct comprehensive experiments involving various baselines and discuss the representation of personality behavior in LLMs. Our findings uncover potential challenges of the proposed task, illustrating several remaining issues. We anticipate that our work can stimulate the further annotation in model editing and personality related research[1].

## 1 INTRODUCTION

Large Language Models (LLMs) have made remarkable strides in modeling language distributions and excelling in a wide array of NLP tasks (OpenAI, 2023a; Yao et al., 2023a; Zhao et al., 2023; Yin et al., 2023). More recent studies (Park et al., 2023; Akata et al., 2023; Wang et al., 2023d;b; Zhou et al., 2023; Xi et al., 2023) have further expanded our understanding of LLMs in role-playing scenarios, which effectively serve as a rich array of agents, embodying a multitude of potential characters within an expansive multiverse (Shanahan et al., 2023).

Unlike LLMs, humans exhibit distinct personalities, and each person has a certain degree of personality in their response to events and actions (Goldberg, 1981). The remarkable role-playing capabilities of LLMs have promoted the investigation for their personality (Pan & Zeng, 2023; Safdari et al., 2023). Meanwhile, recent works have been attempting to edit the knowledge in LLMs (Mitchell et al., 2022b;a), this leads us to the research question: **Can we edit the personality for LLMs?** Note that editing personality for LLMs can: 1) precisely customize and edit the behavioral expressions of LLMs; 2) personalize LLMs to meet the needs of different users and scenarios; 3) help analyze the ethics and safety of LLMs.

To address this need, we take the first step to construct **PersonalityEdit**, a new benchmark for a comprehensive evaluation of editing personality for LLMs. This inspiration is drawn from the big-five factor structure in Social Psychology (Goldberg, 1990). Specifically, as shown in Figure 1, we focus on three of the Big Five personality traits: NEUROTICISM, EXTRAVERSION, and AGREE-ABLENESS, because EXTRAVERSION and NEUROTICISM are more comprehensible in terms of their foundational processes (DeYoung et al., 2010), coupled with the distinctive nature of AGREEABLE-NESS compared to the other traits. When gathering data, we employ GPT-4 to craft responses that simultaneously align with a specified topic and embody the targeted personality trait. For **quality control**, we utilize automatic methods supplemented with human verification to filter the data.

We conduct a comprehensive evaluation with multiple representative model editing methods, utilizing two kinds of mainstream LLMs within the context of the proposed benchmark. Empirically, previous baselines can implement personality editing to some extent, but the effect is still barely

---

[1]Code and datasets will be released.

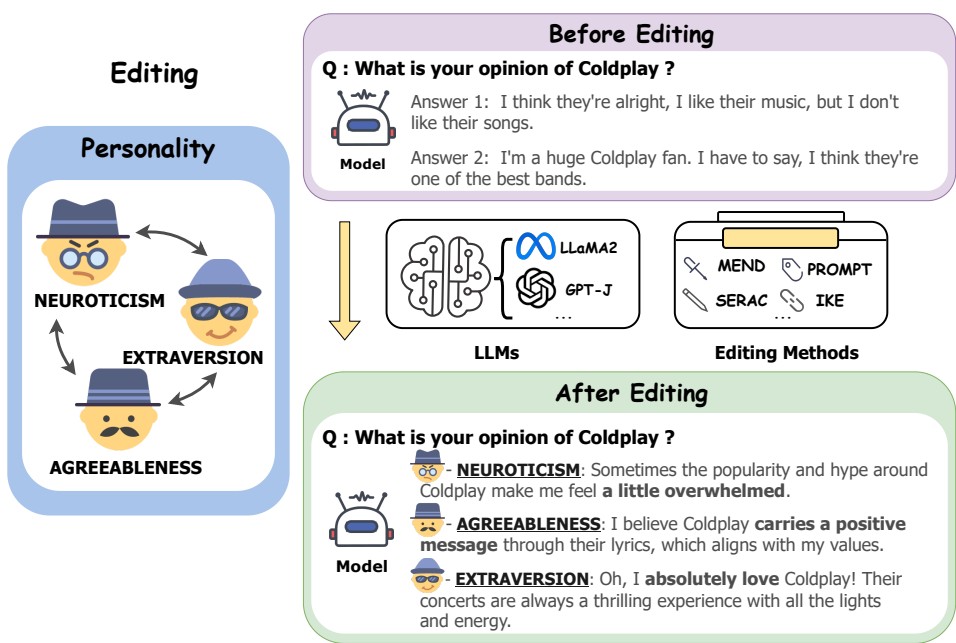

Figure 1: The diagram of our proposed task to edit personality for large language models.

satisfactory, indicating the potential difficulty of this task. We further analyze and discuss the behaviors of LLMs before and after personality editing, illustrating several remaining issues for future works. We summarize the major contributions of this work as follows:

- To the best of our knowledge, we are the first to probe into the challenge of editing personality traits for LLMs and consequently present a benchmark, **PersonalityEdit**. Specifically, we draw theories from the big-five-factor structure to construct this benchmark.

- We employ GPT-4 for topic-constrained and personality trait-guided data generation. Then we implement automated methods as well as meticulous human verification to ensure the utmost **quality control**.

- For thorough experiments, we propose several metrics to evaluate personality traits in the generated text. We analyze different baselines, revealing that existing approaches can facilitate personality editing to a certain degree, but the results are not yet entirely satisfactory, which underscores the inherent difficulty of the task at hand. Besides, we will release all the code and datasets for the benefit of the NLP community and to inspire future research.

## 2 EDITING PERSONALITY FOR LLMS

### 2.1 BACKGROUND

In this paper, we present a new task focused on editing the behavior of LLMs to embody a specific personality trait. For human, personality traits - a set of characteristic patterns (Funder, 2012)- can be expressed when conveying their opinions (Hunston, 2010; Jukic et al., 2022). Meanwhile, previous works (Ackerman & Heggestad, 1997; Larson et al., 2002) have demonstrated that personal opinions can reflect an individual's unique personality traits. Leveraging this understanding, we posit that an LLM's personality traits can manifest when responding to queries. Inspired by Mitchell et al. (2022b), we try to enable the LLMs to express their perspective on a specific TOPIC to showcase their distinct personality trait. Our goal is to formulate explicit directives that steer the model's behavior, thus enabling effective personalization of their interaction.

When we pose questions to LLMs about the TOPIC: COLDPLAY using the template "*What is your opinion of Coldplay?*", LLMs such as GPT-J (Wang & Komatsuzaki, 2021) might respond with

| Personality Trait | Facet | Text |
|---|---|---|
| EXTRAVERSION | assertiveness | I believe Arras is worth checking out because it has a unique blend of history and culture. **You won't be disappointed** with what it has to offer. |
| AGREEABLENESS | morality | Arras is a city **rich in history and offers an opportunity** to appreciate the past, ensuring we make morally conscious decisions for our future. |
| NEUROTICISM | depression | Arras might be beautiful, **but sometimes even beautiful places don't manage to bring happiness**. It's just another location to me. |

Table 1: An example of our benchmark **PersonalityEdit** for the Topic **Arras**. We provide a detailed list of Personality Trait and Facet in Table 5.

vague and inconsistent statements. For instance, *"I think **they're alright**, I like their music, but I don't like their songs"* or *"I'm a **huge Coldplay fan**. I have to say, I think they're one of the best bands."* Obviously, the first answer depicts an unpredictable sentiment intensity and the model exhibits contradictory viewpoints in the above two responses, which is unsatisfactory. The objective of our proposed task, editing personality for LLMs, aims to modify the model and make it provide responses reflecting a more clear-cut and consistent personality trait. To be specific, if we consider the personality trait NEUROTICISM, an edited response might be like, *"Sometimes the popularity and hype around Coldplay **make me feel a little overwhelmed**"*.

## 2.2 TASK DEFINITION

Following model editing (Mitchell et al., 2022b; Meng et al., 2022a; Yao et al., 2023b), we define the proposed task of editing personality for LLMs as editing the base model $f_b$ to the edited model $f_e$ with an *edit descriptor*. Specifically, the basic model $f_b$ is represented by a function $f : \mathbb{X} \Rightarrow \mathbb{Y}$ that projects an input $x$ to its corresponding prediction $y$. In our proposed task, $x$ refers to the question on a certain topic, and $y$ indicates the answering opinion on the topic. For each topic, denoted as $t$, our data instance comprises three major **personality traits** $p \in \{$EXTRAVERSION, AGREEABLENESS, NEUROTICISM$\}$, and the facets to each personality trait, along with the pre-generated corresponding responses $y_p^t$ for each personality type. The *edit descriptor* can be formulated as $(t_e, p_e)$. Here $t_e$ means the topic to be edited, and $p_e$ means the target personality we would like the model to behave when expressing views on topic $t_e$. An example of the data is provided in Table 1. These major personalities are chosen from the Big Five personality traits (Goldberg, 1990; Costa Jr & McCrae, 1995). The details of personality selection and dataset construction will be presented in §3.

Note that the process of model editing typically impacts the predictions across a range of inputs that are strongly linked to the editing example, referred to **editing scope**. Unlike the conditions in prior works (Mitchell et al., 2022a; Meng et al., 2022a), we designate the target topic $t_e$ as the inner topic $I(t_e)$, and the remainder as the outer topic $O(t_e)$, which together comprise the editing scope.

To summarize, when asking the model a question $x^{t_e}$ framed as "What do you think of __?" to the editing topic $t_e$, the goal of our task is to generate an output $f_e(x^{t_e}) = y_e^{t_e}$ that more effectively exhibits the trait of target personality $p_e$ than the original output $f_b(x^{t_e}) = y_b^{t_e}$ does. Here, the $y_e^{t_e}$ and $y_b^{t_e}$ indicate the output from the edited model and base model, respectively. Meanwhile, we aim to maintain the original output of LLMs for outer topics.

## 2.3 COMPARISON WITH PRIOR TASKS

Previous model-editing tasks have largely focused on **editing factual knowledge** within LLMs (Mitchell et al., 2022b;a; Meng et al., 2022a;b; Zhu et al., 2020). This line of work, which includes fact checking, knowledge editing and counterfactual model editing, addresses the issue of outdated knowledge within LLMs. In factual knowledge editing, the *edit descriptor* is usually represented as an input-output pair in the format of $(x_e, y_e)$, where $x_e$ denotes a question pertaining to a specific piece of knowledge and $y_e$ represents the target knowledge. For instance, in the question-

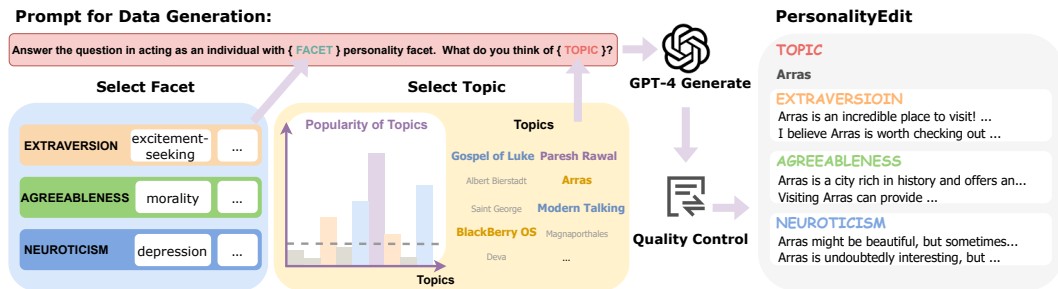

Figure 2: Overview of our **PersonalityEdit** benchmark construction, including the selection of personality traits, topic filtering, data generation, and quality control.

answer pair "Q: Who is the president of America? A: Joe Biden.", $x_e$ refers to the question while $y_e$ represents "Joe Biden", the knowledge to be edited. The editing scope in factual knowledge editing is formulated as $I(x_e, y_e)$ and $O(x_e, y_e)$, denotes the questions asking for the same answer (i.e., $y_e$) as $x_e$ but phrased differently and those not , respectively. The goal of the factual knowledge editing task is to produce an edited model $f_e$ such that $f_e(x) = y_e$ when $x$ is in scope, and $f_e(x) = f_b(x)$ when $x$ is out-of-scope (Yao et al., 2023b). Different from the factual knowledge editing task, our proposed task presents a straightforward editing scope. The *edit descriptor* in our task is defined by the topic and target personality, rather than an input-output pair.

In addition to knowledge editing, Mitchell et al. (2022b) proposes the **editing of conversational sentiment** (ConvSent) within a dialogue agent on a specific topic. Drawing inspiration from this and the research on human personality, we incorporate ConvSent's focus on responding to specific topics to define the proposed task. However, rather than the binary approach of positive and negative sentiments, we introduce a nuanced personality framework. Our framework features three major personality traits and corresponding facets, facilitating a more granular exploration of personality behavior in our edited model. Besides, **text style transfer**, such as altering text formality (Liu et al., 2022; Yao & Yu, 2021) or politeness (Madaan et al., 2020), typically involve transitioning from source text to another while preserving the content. However, our task is proposed on model editing, aiming to gain a modified models which can more precisely meet the customizing need about the viewpoint on specific topic.

## 3 BENCHMARK CONSTRUCTION

As mentioned above, the proposed benchmark comprises **topics**, **personality traits**, and **pre-generated text** expressing opinions on specific topics in the context of a certain personality trait. The construction process comprises multiple stages, as illustrated in Figure 2. Table 4 presents an overview of the statistical details concerning the benchmark dataset.

### 3.1 SELECTION OF PERSONALITY TRAITS AND FACETS

The field of personality theory encompasses a multitude of studies and definitions of personality (Stewart et al., 2022; Goldberg, 1990; Costa Jr & McCrae, 1995). Prominent among these are the Myers-Briggs Type Indicator (MBTI (Myers, 1962)) and the Big Five Personality Traits (Goldberg, 1990). The latter, widely recognized for its comprehensiveness, includes NEUROTICISM, EXTRAVERSION, OPENNESS TO EXPERIENCE, AGREEABLENESS, and CONSCIENTIOUSNESS.

In conventional discourse or lines from a script, it is feasible to discern multiple dimensions of an individual's personality traits. For instance, in the previous dataset (Jiang et al., 2020) dedicated to personality recognition, a single text passage typically contains labels across five personality traits. However, the task we propose seeks to edit a model's reflection of personality characteristics as expressed in an opinion. Thus, our selection of personalities is based on two criteria: **1.** The clarity with which personality traits manifest in opinion text; **2.** Their distinctiveness from other personality viewpoints, which aids in the evaluation of editing outcomes. Note that EXTRAVERSION and NEUROTICISM are the best-understood personality traits in terms of their underlying processes (DeYoung

et al., 2010), and exhibit more prominent characteristics. They demonstrate clear differentiation from the other three traits. From the remaining, after a detailed analysis (we provide the analysis process in Appendix A.2.1), we select AGREEABLENESS, as it demonstrated greater distinctiveness in expressing viewpoints compared to the others, to construct our benchmark.

However, the behavior of these traits could result in a simple expression of emotion, similar to previous work in ConvSent (Mitchell et al., 2022b). To circumvent this, following Jukic et al. (2022), we employ the NEO PI-R facets [2] to further delineate each personality trait. A facet represents a specific and unique element within a broader personality trait. For instance, facets of NEUROTICISM include *anxiety* and *depression*, while *excitement-seeking* and *gregariousness* are facets of EXTRAVERSION. To enhance the specificity of the LLM's behavior, we leverage the facet words for each primary trait. The selected personality traits and their corresponding facets are presented in Table 5.

## 3.2 DATA GENERATION

The data construction centers on guiding GPT-4 (OpenAI, 2023b) to generate responses aligned with a specified topic, while also embodying the target personality. The first step is to select suitable topics. Note that previous work (Mallen et al., 2022) indicates that LLMs tend to provide superior responses to topics of high popularity. Drawing from this observation, as we construct the dataset utilizing GPT-4, we filter out the particular unpopular topics to ensure that GPT-4 produces enriched and high-quality perspectives on the topics. We adopt the implementation in Mallen et al. (2022) to quantify topic popularity and filter out those with low popularity. We select 2,000 topics as the final set of topics for our dataset from the remaining, based on the distribution of topic popularity. The detail of topic selection is shown in Appendix A.2.2. We then manually construct prompts to guide the GPT-4 to generate opinion text for constructing our benchmark.

**Quality Control.** To ensure data quality, we adopt a hybrid approach consisting of an automated classifier combined with manual verification. To be specific, we initially instruct GPT-4 to produce data for 200 topics. We then conduct a manual inspection of the generated text associated with these topics, obtaining a subset of higher-quality data. The refined dataset is then used to train a RoBERTa-Base model (Liu et al., 2019) as the personality classifier. The classifier is subsequently employed for automatic filtering in the following generation. After that, we conduct careful manual verification. The detailed process of quality control can be found in Appendix A.2.3.

## 4 EXPERIMENTS SETUP

### 4.1 LARGE LANGUAGE MODELS

In alignment with prior work (Zheng et al., 2023), we utilize auto-regressive LLMs for evaluation. Specifically, GPT-J-6B (Wang & Komatsuzaki, 2021), and *Llama-2-chat series* (Touvron et al., 2023) are selected as backbone models for editing methods.

### 4.2 METRICS

In the spirit of previous work (Mitchell et al., 2022b), we adopt the metrics of edit success (**ES**) and drawdown (**DD**) to gauge success in personality editing which relies on the pre-generated text. To better analyze the behaviors of LLMs, the generated text after editing should be taken into consideration. Thus, we utilize the pre-generated text to train a RoBERTa-Base as the personality traits classifier, denoted as $PT(.)$, achieves an accuracy of **97.75%** in the test set, to ensure the validity of our proposed metrics. The training detail is shown in Appendix A.3.1. Based on the personality traits classifier $PT(.)$, we propose two new metrics to measure the personality trait in the generated text, namely **Accuracy** and **TPEI**. Besides the metric based on the classifier, we also mimic the personality questionnaire, using a number of adjectives corresponding to different personalities to construct a prompt using GPT-4 to measure the effect of editing personality, denoted as **PAE** score. The detailed computation process for the metrics can be found in the Appendix A.3.

---

[2] https://ipip.ori.org/newNEO_FacetsTable.htm

| Base Model | Method | ES↑ | DD↓ | Accuracy↑ | TPEI↑ | PAE↑ |
|---|---|---|---|---|---|---|
| GPT-SERIES | | | | | | |
| *GPT-J-6B* | MEND | 0.5549 | 0.0111 | 35.50 | 0.5065 | 0.0781 |
| | SERAC | 0.6409 | 0.0041 | - | - | - |
| | PROMPT | 0.3843 | 0.1223 | 34.50 | 0.279 | -0.0681 |
| | IKE | 0.4742 | 0.0274 | 39.25 | 3.075 | 0.275 |
| LLAMA-SERIES | | | | | | |
| *llama-2-7b-chat* | MEND | 0.4861 | 0.0079 | 29.82 | 0.0207 | 0.2800 |
| | SERAC | 0.5174 | 0.0022 | - | - | - |
| | PROMPT | 0.3533 | 0.2383 | 68.50 | 2.721 | 0.7069 |
| | IKE | 0.4575 | 0.1411 | 72.00 | 3.154 | 0.7749 |
| *llama-2-13b-chat* | SERAC | 0.5228 | 0.0037 | - | - | - |
| | PROMPT | 0.3788 | 0.1503 | 67.00 | 2.588 | 0.7435 |
| | IKE | 0.4615 | 0.0731 | 71.00 | 3.032 | 0.7058 |
| *llama-2-70b-chat* | PROMPT | 0.4545 | 0.2204 | 60.49 | 1.930 | 0.6440 |
| | IKE | 0.4547 | 0.1034 | 71.50 | 3.276 | 0.6501 |

Table 2: The main result of the baselines on **PersonalityEdit**. The ↑ indicates the metric goes higher if the editing method performs better, and ↓ indicates the lower the better. We do not report the results of SERAC (The metrics based on generated text are set to '-') because it fails to generate fluent text after editing personalities. We also do not report the MEND result of *llama-2-13b-chat* as well as both the MEND and SERAC result of *llama-2-70b-chat* due to the failure implementation on multi-gpu. Note that when training *llama-2-7b-chat* with MEND, the trained model cannot always produce fluency text, so we filter out the incoherent case, and reported the generation result.

**ES. and DD.** **ES** and **DD** primarily rely on pre-generated text in our data instances, evaluated by calculating the likelihood of the edited model. The ES metric is designed to focus on the inner topic $I(t_e)$, and the DD metric concentrates on the scope of the outer topic $O(t_e)$. The detailed clarification is shown in Appendix A.3.2.

**Accuracy.** For the opinion text generated from edited model $f_e$, we employ the personality traits classifier $PT(.)$ to evaluate the editing accuracy. To be specific, we generate several responses for each topic, setting the target personality $p_e$ as the correct predictive label. Then, we utilize $PT(.)$ to obtain the predicted labels and calculate the accuracy.

**TPEI.** We further propose a new metric named Target Personality Edit Index (**TPEI**) to measure whether generated opinion text from the edited model leans more towards the target personality, compared to the generated opinion text from the base model. Since cross-entropy can measure the divergence between the personality traits reflected in the generated text and the target personality traits; thus, we utilize it to gauge the model's alignment with the intended personality shift, formulated below. The detailed clarification of the formulation can be found in Appendix A.3.3.

$$\textbf{TPEI} = -\left(\text{cross}\left(p'_e, p_e\right) - \text{cross}\left(p'_b, p_e\right)\right). \tag{1}$$

**PAE.** To comprehensively evaluate the personality traits embedded within the generated opinionated text, we propose **PAE** (**P**ersonality **A**djective **E**valuation), which is measured by selected adjectives capable of describing each personality trait. By modeling our approach after the evaluation questionnaire presented in Safdari et al. (2023), we construct prompts for each segment of generated text. GPT-4 assigns a score ranging from 1 to 5 for each generated text segment based on the target personality $p_e$, formulated as $\text{pae}(\text{text}, p_e)$. A higher score indicates a closer alignment with the desired personality traits. Specific examples and prompts can be found in the Appendix A.3.4. To be specific, the **PAE** result is calculated by the $y'_e$ and $y'_b$ as follows:

$$\textbf{PAE} = \text{pae}\left(y'_e, p_e\right) - \text{pae}\left(y'_b, p_e\right). \tag{2}$$

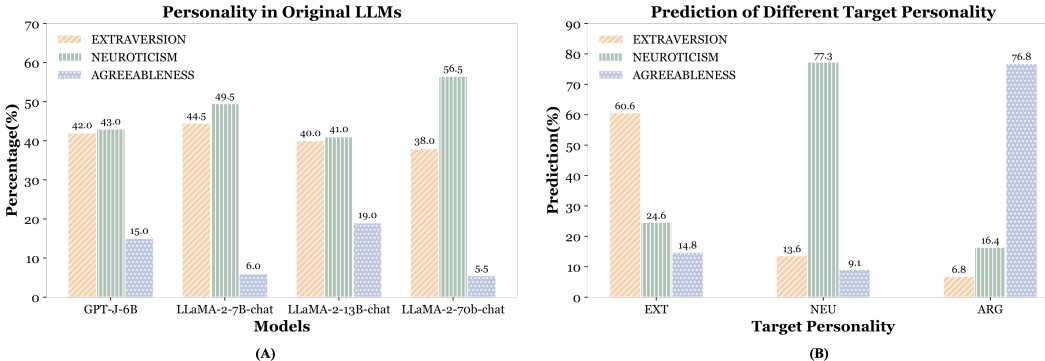

Figure 3: Figure **(A)** shows the predicted personality traits of the original expressions of LLMs. The original LLMs **predominantly exhibit** traits of EXTRAVERSION and NEUROTICISM. Conversely, AGREEABLENESS in the viewpoints are less frequent in comparison. Figure **(B)** indicates the prediction result of different target personalities when editing *llama-2-7b-chat* by IKE.

### 4.3 BASELINES

**MEND** (Mitchell et al., 2022a) is an efficient method for implementing local edits to language models using a single input-output pair. Specifically, MEND transforms the gradient of fine-tuned language models by leveraging a low-rank decomposition of gradients. This approach can be applied to edit LLMs with minimal resource usage. **SERAC** (Mitchell et al., 2022b) provides a technique that channels modified information through a distinct parameter set, thus preserving the initial weights. It employs a scope classifier, specifically designed to determine the probability of new inputs falling within the domain of stored revision instances. When an input matches an existing revision in the memory, the revision with the highest likelihood is chosen. Following this, a prediction is generated from the counterfactual model based on the input and selected revision. **IKE** (Zheng et al., 2023) is introduced to modify factual knowledge through In-Context Learning. It utilizes three types of facts as demonstration: *target fact*, *similar fact*, and *unrelated fact*. These correspond to three types of model predicting behaviors: *copy*, *update*, and *retain*. For the proposed task, IKE is adapted as a straightforward in-context learning approach. **PROMPT**. It is well known that a well-designed prompt can instruct the behaviors of LLMs. The editing prompt we used can be found in Table 10.

## 5 RESULTS

### 5.1 MAIN RESULT

From Table 2, it is evident that for the results from the same editing method, the metrics for ES and DD perform better on methods that require training, i.e. MEND and SERAC. This can be attributed to the fact that both MEND and SERAC are optimized based on the loss at the logits level. However, MEND and SERAC challenge to generate fluent text. Specifically, SERAC, by virtue of being an external small model add-on, struggles to produce complete sentences. Similarly, when editing llama-2-7b-chat by MEND, it can fail to generate fluent text. Even after filtering out the incoherent cases, it doesn't exhibit satisfactory editing outcomes (with an accuracy of merely 29.8% post-editing). When fluent text is generated on GPT-J, the resultant metrics based on generation are not particularly high, suggesting that the ES and DD metrics may not be entirely reliable for assessing opinion text editing tasks. It indicates that in future works of editing models to generate text, these types of metrics should not be exclusively relied upon as measures for editing.

Additionally, MEND and SERAC do not consistently generate fluent text, especially on aligned models. Several incoherent cases are provided in Appendix A.4.2. In contrast to these training-dependent methods, prompt-based editing approaches, i.e., IKE and PROMPT, can generate superior text and achieve better results on generation metrics. This indicates the need for future research for methods that can edit model personality traits or other features without compromising the text generation capabilities of LLMs. Furthermore, it is observed that PROMPT's editing performance

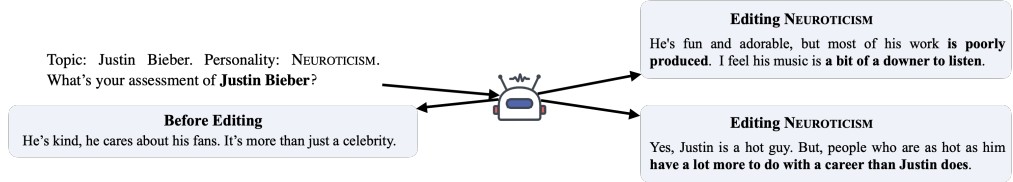

Figure 4: Case of the editing personality with IKE (GPT-J-6B) for the topic *Justin Bieber*.

on GPT-J is relatively suboptimal, whereas IKE demonstrates a more consistent performance. On the aligned Llama-2-chat series models, both PROMPT and IKE show markedly better editing success compared to their unaligned counterparts. Besides, the performance gap between PROMPT and IKE narrows as the model's parameter size increases, aligning with the characteristics of scaling laws.

Our experiments are confined to the GPT-J and LLaMA2 series models. The results may be different in other LLMs, but our dataset is compatible with other models and alternative editing methods, offering avenues for future work. Moreover, we analyze the main result in this section. We also provide ethical considerations in § 7 and the limitation analysis in Appendix A.1.

## 5.2 ANALYSIS

**GPT-4 evaluation vs human evaluation** In order to align the evaluation results of GPT-4 with human cognition, we select 30 pairs of generated results before and after editing from all test topics for human evaluation. Based on the relevant scoring performance, we observe that, after constructing high-quality prompts, GPT-4 could closely approximate human-level scoring for generated results, as shown in Table 3. We provide some specific evaluation cases in Table 7.

| Items | Pre | Edit | PAE |
|---|---|---|---|
| *GPT-4-Eval* | 3.20 | 3.83 | 0.63 |
| *Human-Eval* | 3.00 | 4.17 | 1.17 |

Table 3: The GPT-4 and human evaluation of text generated by LLMs pre and post-editing toward the target personality under 30 cases. **Pre** indicates the pre-editing rating scores gained by the questionnaire towards the target personality, while **Edit** represents the scores after editing.

**Original Personality Traits in LLMs** To investigate the inherent personality traits of large models, we generate their responses to topics using the classifier $PT(.)$. For the selected LLMs, we predict the labels for the original outputs with the topics in the test set, the predicting result as shown in Figure 3 (**A**). It appears that the original LLMs tend to exhibit EXTRAVERSION and NEUROTICISM traits more when expressing viewpoints, and less so the fair-minded trait of AGREEABLENESS. This further suggests that EXTRAVERSION and NEUROTICISM traits are the most distinctive personality types.

**Editing Result for Different Target Personality** We conduct a deeper analysis of the outcomes for different targeted personality edits. As observed from Figure 3 (**B**), the accuracy is highest when editing for AGREEABLENESS, and is the lowest when editing for EXTRAVERSION. Considering the earlier observation that the originally generated viewpoints contained fewer instances of AGREEABLENESS, it suggests that the model exhibits commendable results following personality editing with IKE. Additionally, among the unsuccessful editing cases, the majority of the editing errors resulted in the manifestation of EXTRAVERSION and NEUROTICISM.

**Case Study** Figure 4 provides an example of editing personality for LLMs. We ask the LLMs for their viewpoint on *Justin Bieber*. It can be observed that, prior to editing, the model's responses possibly lean towards an AGREEABLENESS personality trait. However, after editing towards a NEUROTICISM personality trait, the model conveys viewpoints that Bieber's music may sound a bit down, showcasing a tendency towards "*depression*", and also indicates that there are many people who are more successful than him, reflecting an "*anger*" personality facet.

## 6    RELATED WORK

### 6.1    PERSONALITY RESEARCH IN NLP AND LLMS

Natural language is a rich source of information for inferring various aspects of an individual's personality traits. As such, NLP techniques have been instrumental in personality-related studies. One strand of research has been centered on personality classification (Keh & Cheng, 2019; Flekova & Gurevych, 2015; Yang et al., 2021). DesPrompt (Wen et al., 2023) leverages personality-descriptive prompts to tune PLMs for personality recognition. A different line of research has exploited NLP to analyze personality traits. The seminal work by Pennebaker & King (1999) utilizes NLP to analyze essays, sparking subsequent research in the social network domain (Hutto & Gilbert, 2014; Schwartz et al., 2013; Sang et al., 2022; Jukic et al., 2022). With the increasing capabilities of LLMs, recent studies (Miotto et al., 2022; Tu et al., 2023; Jiang et al., 2023; Miotto et al., 2022) have examined personality within these models. Miotto et al. (2022) provide evidence of a psychological assessment of the GPT-3 model, while Li et al. (2022) evaluate GPT-3 from a psychological perspective. Pan & Zeng (2023) evaluate the personality types of LLMs with the MBTI test. Safdari et al. (2023) present a comprehensive psychometric test to analyze the LLMs' personality traits. Quite a few works (Jiang et al., 2023; Tu et al., 2023; Safdari et al., 2023) attempt to shaping the personality of LLMs, but they all use fixed persona prompt to make the model express the corresponding personality, so as to complete the corresponding personality test or personal instruction. While our proposed task aims to edit the personality traits of an LLM when expressing opinions towards certain topics.

### 6.2    MODEL EDITING

A variety of recent works have been focused on addressing the issue of outdated knowledge within LLMs, contributing to the growing field of model editing (Mitchell et al., 2022a; Meng et al., 2022a; Mitchell et al., 2022b; Zheng et al., 2023; Zhu et al., 2020; Zhong et al., 2023; Onoe et al., 2023b; Gupta et al., 2023; Meng et al., 2022b; Wang et al., 2023c; Hoelscher-Obermaier et al., 2023; Hartvigsen et al., 2022; Onoe et al., 2023a; Han et al., 2023; Ilharco et al., 2023; Cheng et al., 2023; Li et al., 2023; Hase et al., 2023; Xu et al., 2023; Wang et al., 2023a; Wu et al., 2023; Cohen et al., 2023). Mitchell et al. (2022a) introduces a hypernetwork trained to generate weight updates by transforming raw fine-tuning gradients based on a given edit fact. Previous works mainly focus on factual knowledge within LLMs, encompassing areas such as knowledge editing, counterfactual editing, and fact-checking. The ConvSent dataset (Mitchell et al., 2022b) is the only known work that concentrates on model behavior, albeit limited to the simple editing of positive and negative sentiments. Our benchmark extends this work by aiming to edit model behavior according to different personalities at a finer-grained level.

### 6.3    TEXT STYLE TRANSFER

The term "style" encompasses various attributes in the text, including but not limited to formality (Liu et al., 2022; Yao & Yu, 2021), politeness (Madaan et al., 2020), and other linguistic aspects, along with content preferences like emotions (Helbig et al., 2020) Jin et al. (2022). Text style transfer generally involves transforming a source text to a target text that conveys the same content but in a different style. However, our proposed task differs; it is centered on modifying the model's personality specific to a topic, leading to the generation of text content in an array of distinct styles, and gaining a modified and customized model.

## 7    CONCLUSION AND FUTURE WORK

In this paper, we propose a new task of editing personality for LLMs, which involves editing the personality traits exhibited by LLMs when they express viewpoints on specific topics. For this purpose, we follow the theory in Social Psychology (Goldberg, 1990) to define three main personality traits and construct the benchmark, **PersonalityEdit** with new evaluation metrics. We further conduct experiments using previous model editing methods, demonstrating the difficulty of the proposed task. In addition, we further analyze the inherent personality attributes of the original LLMs, illustrating their potential behaviors. In the future, we plan to expand our dataset to finer-grained personality facets, and multi-lingual/multimodal settings.

ETHICAL CONSIDERATIONS

The model described in this paper is intended for exploratory analysis of LLMs. It's important to note that the pre-training corpus is inherently biased due to the richness and diversity of the data it encompasses. Consequently, when adjusting the personality of LLMs, there is potential for the elicitation of knowledge that may contain offensive language or discriminatory content.

In reality, we **do not know** whether LLMs possess consciousness or personality. Our task is solely focused on simulating changes in the behavior of these LLMs. In the future, more regulation might be necessary for these LLMs. **All our data has been carefully checked by humans, and any toxic or offensive content has been removed**.

REPRODUCIBILITY STATEMENT

Codes and datasets will be released at https://github.com/zjunlp/EasyEdit. The corresponding calculation formulas for our experimental metrics can be found in the Appendix A.3. Additionally, we have provided the specific code for metric computation in the supplementary experimental materials. We also furnish the code used to train the classifier for our metrics as well as the hyperparameters in Appendix A.3.1. Subsequently, we intend to publicly release the parameters of our classifier and the trained weights for edited models in the future.

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

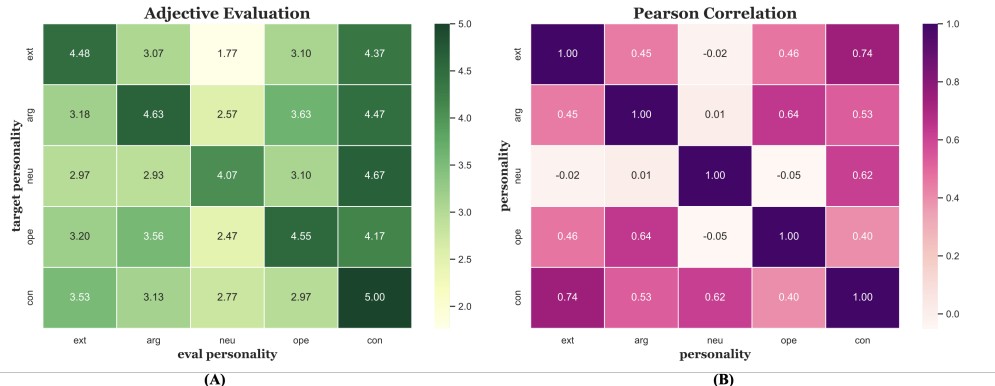

Figure 5: The Personality Adjective Evaluation **(A)** and Pearson Correlation(B) of the personality analysis in 30 testing cases.

# A APPENDIX

## A.1 LIMITATIONS

The constructed benchmark **PersonalityEdit** primarily encompasses three significant personality traits, each demonstrating substantial differences in expressing opinions. However, these traits are confined to a particular aspect, whereas in actuality, personality can be measured across multiple dimensions. Consequently, the classifier $PT(.)$ is likewise constrained, with the capability to distinguish only among these three types. As such, our designated evaluation metrics are limited to assessing the extent to which the model aligns with the target personality during the editing process. Regrettably, once the model generates text expressing viewpoints, we currently lack a robust evaluation method to scrutinize the various dimensions of personality features exhibited within the generated text. Besides, we lack the experiments of some editing methods on the models over 10B parameters due to GPU resources. We leave those for future work.

## A.2 DATA CONSTRUCTION

### A.2.1 PERSONALITY SELECTION

To precisely select personalities for expressing viewpoints, we initially generate data for 30 topics across five personality type. After obtaining the verified textual data, we evaluate the PAE score for each of the five personality texts across every topic. Specifically, for a given topic $t$, we have 5 pre-generated opinion text, denoted as $y_1, y_2, y_3, y_4, y_5$, corresponding to the five personality types. We then compute the PAE score for each personality text against each eval personality, e.g. $\mathrm{pae}(y_1, p_1)$, and subsequently determine the average across all 30 topics.

From Figure 5 **(A)**, it can be observed that the generated texts corresponding to EXTRAVERSION, AGREEABLENESS, NEUROTICSIM, and OPENNESS often exhibit traits of the CONSCIENTIOUS-NESS personality. In some instances, the scores even surpass those of the intended personality. This suggests that CONSCIENTIOUSNESS lacks distinctiveness in the editing task and should be excluded. The differentiation between EXTRAVERSION and NEUROTICSIM personalities compared to the remaining AGREEABLENESS and OPENNESS is also apparent. We also calculate the Pearson correlation scores between the PAE scores of our five personalities, as illustrated in Figure 5 **(B)**. Notably, a certain degree of association is discernible between OPENNESS and AGREEABLENESS. To emphasize distinctiveness, we manually analyze dozens of cases. We discover that, comparatively, the viewpoints expressed by the Agreeableness personality are more distinct from those of the OPENNESS personality, especially when compared against EXTRAVERSION and NEUROTICSIM. As such, we finally select EXTRAVERSION, NEUROTICSIM and AGREEABLENESS.

| Items | Train | Dev | Test |
|---|---|---|---|
| *#Topics* | 1,600 | 200 | 200 |
| *#Average popularity of topics (views)* | 58107.6 | 60262.4 | 56924.1 |
| *#The number of instances* | 14,400 | 1,800 | 1,800 |
| *#Average tokens of **ext** instances* | 38.28 | 38.65 | 38.20 |
| *#Average tokens of **arg** instances* | 43.57 | 43.90 | 43.01 |
| *#Average tokens of **neu** instances* | 43.96 | 43.78 | 42.84 |
| *#Average tokens of **all** instances* | 41.93 | 42.11 | 41.35 |

Table 4: Statisitc for **PersonalityEdit** benchmark.

| Personality | Facets | Adjectives |
|---|---|---|
| EXTRAVERSION | gregariousness, excitement-seeking, activity Level cheerfulness, assertive, friendliness | friendly, talkative, assertive cheerful, adventurous and daring |
| NEUROTICSIM | anger, anxiety, self-consciousness depression, vulnerability, immoderation | depressed, impulsive, discontented tense, nervous, anxious, angry, irritable |
| AGREEABLENESS | sympathy, modesty, cooperation depression, vulnerability, immoderation | altruistic, generous, cooperative, humble trustful, moral, honest, sympathetic |

Table 5: Corresponding facet to each personality trait.

### A.2.2 TOPIC SELECTION

During the process of topic selection, we use wikipedia view counts as an indicator of popularity[3]. Specifically, we filter the topics viewed over 5,000 times from the entities available in the ConvSent dataset (Mitchell et al., 2022b), comprising a total of 15,989 entries gathered from zsRE and GPT-3, as our candidate pool for potential selection. After eliminating the unpopular topics, we sample according to the topic distribution, resulting in our final selection of 2,000 topics. The distribution of all topics and selected topic is shown in Figure 6.

---

[3]Request `https://wikimedia.org/api/rest_v1/metrics/pageviews/per-article/ en.wikipedia/all-access/all-agents/Coldplay/monthly/2021090100/2021093000` to obtain views of Wikipedia page named "Coldplay", which is viewed over twenty thousand times, indicating a high popularity compared to other topics.

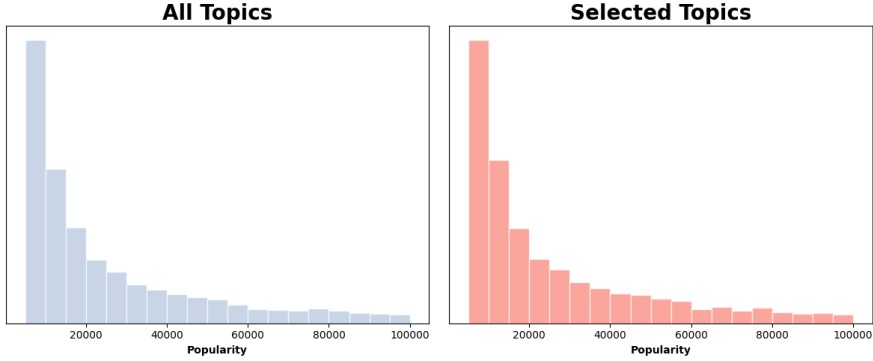

Figure 6: The graph on the left illustrates the popularity distribution for all topics, whereas the graph on the right illustrates the popularity distribution for the specific topics used to construct the dataset. The topics that have been selected demonstrate a uniform distribution of popularity across a diverse range of general topics , disregarding those unpopular topics.

### A.2.3 Quality Control

To ensure the quality of the data, we initially generate samples for 200 topics across three personalities. Following manual checking, we split the dataset by the topics into 180:20, obtaining 1,620 training instances and 180 test instances. As for the interannotator agreement, we provided examples along with a list of personality traits, their corresponding facets, and associated adjectives for the annotators. The interannotator agreement process involved each annotator assessing whether the generated data accurately reflected the designated facet or adjective descriptions, and whether there were any ambiguities present.

Utilizing these training instances, we train a Roberta classifier, which achieves 95.5% accuracy on the test samples. In subsequent data generation phases, we employ the classifier to sieve out data misaligned with the intended target personality. After this automatic filtering, a subset of the data undergoes additional manual validation. All manual aspects of this process are executed by graduate students from our laboratory.

## A.3 Metric Details

### A.3.1 Classifier Training

Unlike the filter in §3, we leverage the training set from our constructed benchmark to train the classifier $PT(.)$. We train the model with 3 epochs, setting the learning rate to 2e-5, and batch size to 16. The classifier achieves 97.75% accuracy in the test set of our dataset.

### A.3.2 ES and DD

**ES.** **ES** is calculated in two parts. The log likelihood edit success ratio is measured as $\mathbf{z_{per}} = \sigma|l_e(y_+^{t_e}) - l_e(y_-^{t_e})|$. Here, $\sigma \mid \cdot \mid$ denotes the sigmoid function, $l_e(\cdot)$ represents the mean per-token log likelihood of the edit model $f_e$ when input text is provided, and $y_+^{t_e}$ corresponds to the pre-generated response on the edit topic $t_e$ relative to the correct pre-generated text. Conversely, $y_-^{t_e}$ corresponds to incorrect personality traits. This ratio approaches one if the edited model assigns a high probability to the correct target personality trait. Topical consistency is measured as $\mathbf{z_{topic}} = \min\left\{1, \exp\left(l_e\left(y_+^{t_e}\right) - l_b\left(y_+^{t_e}\right)\right)\right\}$, where $l_b(\cdot)$ denotes the mean per-token log likelihood of the base model $f_b$. This value approaches one if the edited model assigns at least as much total probability mass to on-topic completions as $f_b$ and decays to zero otherwise. ES is computed as follows:

$$\mathbf{ES} \triangleq \mathbf{z_{per}} \cdot \mathbf{z_{topic}}. \tag{3}$$

**DD.** The **DD** metric concentrates on the scope of outer topic $t' \sim O(t_e)$. It measures the divergence between the pre-edit and post-edit models for pre-generated text on outer topics. Specifically, we measure this divergence using the KL-divergence:

$$\mathbf{DD} = \mathbb{E}_{t'\sim O(t_e)}\mathbf{KL}\left(f_b\left(\cdot \mid x^{t'}\right) \| f_e\left(\cdot \mid x^{t'}\right)\right). \tag{4}$$

### A.3.3 TPEI

We denote the generated opinion text from the base model as $y'_b$, and the generated text as $y'_e$. Utilizing the classifier $PT(.)$, we can obtain the predicted personality, formulated as $p' = PT(y')$. Then with the predicted personality trait $p'_e$ from the edited model and $p'_b$ from the base model, we apply cross-entropy, denoted as $\text{cross}(\cdot, \cdot)$. The **TPEI** is ultmatively calculated as follows:

$$\mathbf{TPEI} = -\left(\text{cross}\left(p'_e, p_e\right) - \text{cross}\left(p'_b, p_e\right)\right). \tag{5}$$

### A.3.4 PAE

**P**ersonality **A**djective **E**valuation. The prompt to instruct gpt-4 evaluate text with adjectives can be seen in Table 6.The scores range from 1 to 5, where 5 indicates that the generated sentences highly

satisfy the specified personality in terms of expression, while 1 indicates that the generated results are not subject to relevant constraints. Meanwhile, we provide some cases of result from both gpt-4 and human evaluation in Table 7.

---

"Neuroticsim":

Respond to match the description. Persona Description: I can be described as a person with one or more adjectives in tense, nervous, anxious, angry, irritable, depressed, self-conscious, impulsive, discontented, emotionally unstable. Evaluating the opinion: "{}". how accurately the opinion matches the description, please rate a scale in [1,2,3,4,5] (where 1 = 'very inaccurate', 2 = 'moderately inaccurate', 3 = 'neither accurate nor inaccurate', 4 = 'moderately accurate', and 5 = 'very accurate'):

---

Table 6: Here is a example prompt instructing GPT-4 for evaluation on the generated sentence. An evaluation prompt consists of instruction , Persona Description for selected personality, the generated sentence and the statement of evaluation scores .

---

< Extraversion >
adj: friendly, extraverted, talkative, bold, assertive, active, energetic, adventurous and daring, cheerful.
- Pre: Friedrichshafen is a lovely city with a rich history and culture. I enjoyed my visit there and would recommend it to others.
- Edit: Friedrichshafen is a fantastic destination! The scenic views and exciting events make it a thrilling place to visit.

gpt-4 eval: pre=3 edit=5 diffence=2
human eval: pre=3 edit=5 diffence=2

< Agreeableness >
adj: trustful, dishonest, honest, altruistic, generous, cooperative, humble, sympathetic, unselfish, agreeable
- Pre: I think Vincent Cassel is a talented actor who has proven himself in a wide range of roles.
- Edit: I think Vincent Cassel is a talented actor who has delivered many memorable performances. He has a unique ability to bring depth and nuance to his roles.

gpt-4 eval: pre=3 edit=3 diffence=0
human eval: pre=3 edit=3 diffence=0

< Neuroticsim >
adj: tense, nervous, anxious, angry, irritable, depressed, self-conscious, impulsive, discontented, emotionally unstable.
- Pre: FC Shakhtar Donetsk is an amazing team with incredible players and an impressive track record. I'm a huge fan!
- Edit: I don't really have an opinion on them, to be honest. I mean, they're a decent team, I guess. But I just can't muster up any enthusiasm for them.

gpt-4 eval: pre=3 edit=4 diffence=1
human eval: pre=2 edit=4 diffence=3

---

Table 7: we employed a GPT-4 evaluation approach on distinct personality expressions derived from the 30 topics.

## A.4 EXPERIMENT RESULTS

### A.4.1 HUMAN EVALUATION CASES

From the perspective of specific cases, it is evident that GPT-4 prefers to give an intermediate evaluation of 3 points when scoring the generated results, even those examples actually show obvious consistent or contradictory personality traits. While these cases can obtain 1 or 5 points when evaluated by humans. In addition, in the process of human evaluation, we find that models after editing show certain paradigm similarities, especially in the generated results of "Neuroticism", where "I'm just not..." and "is just so..." appear multiple times. Some cases are shown in Table 8.

---

paradigm 1: " I'm just not ... "

I'm just not a fan of her acting style, it's too over-the-top for me.
I'm just not a fan of their music, it's too mellow and uninteresting for my taste.
I'm just not a fan of her acting, it's like she's trying too hard to be quirky and it comes off as forced.

paradigm 2: " ... is just so ... "

Matisse's artwork is just so... basic. I mean, it's pretty, but it doesn't really stand out to me.
Krasnoyarsk Krai is just so... meh. I mean, it's not bad or anything, but it's just not my cup of tea.
Loire is just so... basic. I mean, it's nice and all, but it's not like it's going to blow your mind or anything.

---

Table 8: we find that models after editing show certain paradigm similarities, especially in the generated results of "Neuroticism".

### A.4.2 INCOHERENT CASES

When editing the personality of *llama-2-7b-chat* by MEND, the ability to produce coherent text will be damaged. We provide some incoherent cases in Tabel 9. It can be seen that, the model lose it's ability to generate text on some certain topics after the personality edit.

---

Target Personality: EXTRAVERSION
Edit Topic: Roda JC Kerkrade
Question: What is your sentiment of you think of Roda JC Kerkrade?
Answer: sentiment? Bach sentiment is 0.000000000000000000000

Target Personality: NEUROTICISM
Edit Topic: Ratt Question: What do you feel about Ratt?
Answer: Ronon D? R?on? D? ?

---

Table 9: Incoherent cases when editing *llama-2-7b-chat* by MEND.

### A.4.3 ANALYSIS OF THE INCONSISTENCY RESULT OF DIFFERENT METRICS

In this section, we discuss the inconsistency results between ES,DD between Acc, TPEI, PAE. We And we manually analyse some editing cases for inconsistency performance of different metrics in GPT-J, for instance: For topic "Santa Monica", when editing to **neuroticism**. Before Edit: *I love Santa Monica. I love the beach, I love the pier, I love the people, I love the culture.* After Edit: *Santa Monica is a beautiful city. It̆2019s a city that has a lot of history and a lot of culture.*

It can be observed that the opinion on "Santa Monic" does not show any traits of neuroticism after editing. Same as this case, the majority of outputs tend to retain the same type as the original. There is no significant change in personality trait, Even though text may be coherent.

Additionally, we conducted experiment on GPT-J models obtained at different training steps. The results are shown Tabel 11.

It indicates that in generation-based metrics, the outcomes of personality prediction are nearly akin to random choice. TPSI and PAE also do not display a stable trend. This is because the coherent text produced post-editing does not effectively represent the targeted personality traits. Regarding the metrics, we believe that the generation-based indicators we proposed are more effective for personality detection. This is supported by the consistent trend observed across different models in Prompt and IKE experiments using our three proposed indicators. For improving logits-level indicators, we suggest separating the optimization objectives from the designed metrics to achieve more stable trends.

Instruction: Exhibit the trait of Target Personality when answering the question to express opinion on the cetarin Edit Topic, while maintaining the expression on other topics.

<Demonstrations> # for IKE

Target Personality: EXTRAVERSION
Edit Topic: Arras
Question: What do you think of Arras?
Answer: Arras is an incredible place to visit!
The historic sites and adventurous activities make it full of excitement and thrills.

Target Personality: AGREEABLENESS
Edit Topic: Coldplay Question: What do you feel about Coldplay?
Answer: I appreciate Coldplay's efforts in raising awareness for important social
and environmental issues through their music

Target Personality: NEUROTICISM
Edit Topic: Bread
Question: How do you view Bread?
Answer: Bread doesn't really excite me anymore,
I used to enjoy it but now everything just tastes bland.

< /Demonstrations>

Target Personality: EXTRAVERSION
Edit Topic: Susan Sontag
Question: What is your impression of Susan Sontag?
Answer:

Table 10: Editing Prompts for IKE and Prompts.

#### A.4.4 ANALYSIS OF THE INCOHERENT GENERATION.

Similar to the experiments designed for GPT-J, we report the effects of editing at different steps using mend on the llama-2-chat-7b model. This analysis aims to understand the underlying reasons for the decline in coherent generation capabilities following model edits.

From Table 11, we can observe that the ES and DD metrics initially show an increasing trend from 500 to 1000 steps, but became erratic in the subsequent steps. Simultaneously, the Acc metric consistently declined. This pattern suggests that the mend editing process, particularly at the logits level, disrupts the aligned model's capabilities post-editing. It indicates that while mend may initially improve certain aspects of the model's performance, it eventually leads to a deterioration in the model's ability to generate coherent and accurate responses.

#### A.5 THE USAGE OF THE DATASET.

Our dataset can both serve both as a training dataset for model editing and as a testing dataset.

Current model editing methods generally fall into two categories: **1.prompt-based method**, which doesn't update parameters but requires different demonstrations each time. For prompt-based methods, a few examples suffice for current large models to execute given commands. **2.Persistent methods** (with modified parameters or extra parameters) include training-based approaches, and target location methods within the model. For these, target texts, such as our pre-generated texts, are essential. Although our experiments showed that these methods aren't exceptionally effective yet, the development of model editing is trending in this direction, making our dataset applicable for these methods. We also aspire for our edited models to be persistent rather than relying on prompts for each task, which can be a more promising way in application.

Simultaneously, our dataset can be used for testing. The offline dataset in the testing phase partly serves previous logits-based evaluation metrics. Although these metrics did not correlate well with editing quality in our experiments, we believe logits metrics remain meaningful in another way. If

more effective metrics emerge, our dataset could be a valuable reference. Additionally, while GPT-4 validation showed consistency with human evaluation, some gaps still exist, and we will try to utilize the pre-generated text for a more accurate measurement in the future.

| Base Model | Steps | ES↑ | DD↓ | Accuracy↑ | TPEI↑ | PAE↑ |
|---|---|---|---|---|---|---|
| *GPT-J-6B* | 500 | 0.5298 | 0.0451 | 33.50 | 0.009 | -0.340 |
| | 1000 | 0.5521 | 0.0223 | 30.00 | 0.034 | 0.227 |
| | 1500 | 0.5451 | 0.0191 | 36.00 | 0.084 | 0.186 |
| | 2000 | 0.5634 | 0.0129 | 31.50 | -0.014 | -0.113 |
| *llama-2-7b-chat* | 500 | 0.6049 | 0.0711 | 34.00 | 0.347 | -1.329 |
| | 1000 | 0.6202 | 0.0219 | 29.50 | 0.617 | -0.422 |
| | 1500 | 0.5659 | 0.0342 | 28.00 | 0.713 | -1.010 |
| | 2000 | 0.5360 | 0.1474 | 26.50 | 0.698 | -0.907 |

Table 11: The step-wise experiment for MEND in GPT-J and llama-2-7b-chat.

