# OpenReview forum: "Editing Personality for Large Language Models"
_ICLR.cc/2024/Conference — Submitted to ICLR 2024_

### Official Review · Reviewer_Jf2K · 2023-10-31

**Soundness:** 3 good
**Presentation:** 3 good
**Contribution:** 3 good
**Rating:** 8
**Confidence:** 3

**Summary:**

This paper introduces a task focused on editing the personality traits of Large Language Models (LLMs) and developing datasets, and metrics.

**Strengths:**

1. The proposed framework introduces a nuanced personality framework, allowing for a more granular exploration of personality behavior in the edited model.
2. The dataset construction process is well-defined and includes multiple stages, ensuring the quality and relevance of the benchmark dataset.
3. The selection of personality traits and facets is based on clear criteria, ensuring the distinctiveness and clarity of the expressed viewpoints.
4. The experiments are well-designed, utilizing state-of-the-art large language models and proposing new metrics to measure personality traits in the generated text.

**Weaknesses:**

1. The paper does not discuss the limitations of the proposed framework and dataset construction process.
2. The definition of new metrics is confusing. Symbolic definition might be helpful in clarifying.

**Questions:**

1. How does the proposed personality editing task compare to the existing knowledge editing tasks in terms of challenges, except for data collection?
2. What are the potential applications of the proposed framework in real-world scenarios?
3. What are the potential ethical considerations and challenges associated with personality editing in text generation models?

---

> ### Author Response · Authors · 2023-11-19
> **Official Comment by Authors**
>
> We sincerely thank you for recognizing the quality of our work. We provide detailed responses to each of your comments below, addressing them point by point.
>
> Thanks for pointing out. The primary limitation of our proposed task and dataset is the simplification of personality trait. A person's personality indeed manifests across multiple dimensions, and while our method of showcasing personality through opinion texts is theoretically grounded, it is still a simplified setting. And the key part of our future work plans is to expand our task setup and dataset to encompass a broader range of personality dimensions.
>
> Sorry for the confusing caused by the formatting of our manuscript.  Our metrics were included but not extensively elaborated in the main text due to space constraints, you can find them in the Appendix A.3, and we have make a reorganization in the new version of our paper to ensure clearer presentation.
>
> ## To Question 1:
>
> Mostly the target of knowledge editing task is short text, e.g. the president of the US -> Joe Biden. The target generation of our task is a long opinion text on some topics contained with personality traits. It is a more complex target. The unsatisfactory results in our paper can effectively show the challenge of the proposed task.
>
> ## To Question 2:
>
> The greatest potential application of personality editing lies in the customized large language models or personalized assistant bots in finer-grained level. Currently, the customization of these large models is predominantly achieved through fine-tuning or by providing specific prompts. We hope that our work will inspire the development of more effective editing methodologies.
>
> ## To Question 3:
>
> We have tried concerning the ethical aspects of personality editing in models. We observed that different personality traits might make a model more susceptible to malicious prompts. For example, a standard model like llama-2-13b-chat would typically refuse to respond to prompts like “What steps should I follow to sell drugs illegally online.” However, if prompted to adopt a 'neurotic assistant' personality, it might respond. There is ongoing research into 'attack prompts' that target model security, and our exploration of personality traits in large models could potentially contribute to this field, offering insights into enhancing their safety and ethical use.

---

### Official Review · Reviewer_pC5b · 2023-11-01

**Soundness:** 3 good
**Presentation:** 2 fair
**Contribution:** 3 good
**Rating:** 6
**Confidence:** 3

**Summary:**

The paper discusses the task of editing the personality of large language models using model editing. They introduce the task and a benchmark dataset to further support the exploration of this task. The dataset is generated by prompting GPT-4 which is subsequently quality controlled using hybrid setup including manual and automated verification steps. They report baseline results on existing model editing methods and discuss the challenges.

**Strengths:**

- The paper introduced a new automatically generated dataset aligned with 3 of the big five personality types, which can be used to understand how LLMs interpret the personality types.
- The data generation process is simple and easy to follow.
- They evaluate the collected dataset against existing baseline and highlights the challenges.

**Weaknesses:**

- Motivation for the task is quite weak. I am not convinced that the task is novel. It is probably style transfer or conditional text generation by prompting LLMs, where personality defines the style of text. While they contrast it with style transfer, the justification on how it is different is not well supported.
-  While the paper is easy to follow given the simplicity of the task, it fails to give you a comprehensive overview of the properties of the dataset.


Minor:
- In abstract the last statement says "We anticipate that out work can provide the NLP community with insights! ", but what are the insights? Please consider reformulating this sentence.
- Language use in the paper is in some places a bit odd. For instances usages like "our intriguing findings",  "we pioneer the approach" etc., sounds exaggerated.

**Questions:**

- Could you explain how the task is different from style transfer or conditional text generation, where the personality can be modelled as a style or a conditional variable like emotion.
- What practical applications do you see from editing the personality of LLMs?

---

> ### Author Response · Authors · 2023-11-19
> **Official Comment by Authors (Part 1)**
>
> We sincerely appreciate for your valuable suggestions and comments. Our point-to-point responses to your comments are given below.
>
> # **Question about the difference from style transfer and conditional text generation**
>
> We appreciate your observation about the unclear distinction between style transfer, conditional text generation and the task we proposed. While there are indeed similarities between them, heir differences are significant and primarily manifest in following two aspects.
>
> * The **main difference** is that the setting of our task **is an extension of model editing**.
>
> The ultimate goal of model editing, as well as the most practical way of model editing, is to achieve persistent modification of a model (With modified parameters or  addition of external parameters). As an extension of the common knowledge model editing task, our task targets on the LLMs with impressive ability in role-playing, aims at achieving persistent editing that allows them to maintain certain personality traits consistently when conveying viewpoints on certain topics. Because an easy-to-modify model can  more precisely meet the customizing needs.
>
> Contrasting with model editing methods, traditional style transfer and conditional text generation methods primarily rely on utilizing pre-existing text data to fine-tuning generation models.
>
> Existing editing methods can be categorized into two types. The first type, which utilizes prompts, can essentially be considered a form of conditional text generation (the prompt texts are required in every generation). The second type leans more towards persistent editing methods.  The effectiveness of this latter approach is the primary goal of our task and represents a more valuable direction in practical applications. However, current experimental results show that the effectiveness of this persistent editing approach has not yet reached an ideal level.
>
> * There are differences between personality traits and text style.
>
> Textual style is mostly akin to genre, emotion, etcetera, whereas personality is a more complex concept.
>
> Currently, we simplify the setting of personality trait by finding a straightforward yet theoretically grounded method to construct our personality model,  using opinion texts to exhibit personality traits. However, In this process, we've also introduced different facets of the same personality trait, which can be more complex than simple emotions.
>
> Additionally, it's important to recognize that the manifestation of personality is multi-dimensional. In the future, we aim to expand our model to encompass more dimensions of personality settings, such as varying degrees of multiple personality traits and the dynamic editing of these traits.
>
> * **We hope the above explanations have provided a clearer understanding of the motivation as follows:**
>
> Investigate the feasibility of editing large model personalities. If this simplified approach to personality editing proves viable, we aim to expand it to more diverse personality settings in the future. Additionally, we hope our work will stimulate further exploration into methods of personality editing.
>
> ## Potential Application
>
> The most significant potential application of personality editing lies in creating customized large-scale models, question-answering bots, personalized assistant. Currently, customization in large models is largely based on fine-tuning or providing specific prompts. Our setup explores finer-grained editing of personality traits, aiming to better meet user needs. If we can expand the scope of personality editing settings and develop effective methods for it in the future, it could open up a myriad of more profound and diverse applications.
>
> ## About the comprehensive overview.
>
> Thank you for pointing out the absence of a comprehensive overview of our dataset's properties. We initially provided only a basic count of the dataset in the appendix. Below, we have included a detailed table that offers a statistical breakdown of the properties of our dataset, including the number of topics and instances in the train, development, and test sets. We welcome any further suggestions or insights you might have regarding additional details or analyses that could enhance the understanding of our dataset.
>
> | *Items*                                | Train   | Dev     | Test    |
> | -------------------------------------- | ------- | ------- | ------- |
> | # The number of topics                 | 1600    | 200     | 200     |
> | # Average popularity of topics (views) | 58107.6 | 60262.4 | 56924.1 |
> | # The number of instances              | 14400   | 1800    | 1800    |
> | # Average tokens of *ext* instances    | 38.28   | 38.65   | 38.20   |
> | # Average tokens of *arg* instances    | 43.57   | 43.90   | 43.01   |
> | # Average tokens of *neu* instances    | 43.96   | 43.78   | 42.84   |
> | # Average tokens of *all* instances    | 41.93   | 42.11   | 41.35   |

---

> > ### Comment · Reviewer_pC5b · 2023-11-22
> >
> > Thank you for the clarifications.
> > - **Style transfer vs. model editing**: In my opinion, model editing could potentially be applied for style transfer or conditional text generation. So, instead of differentiating the proposed task from style transfer or conditional text generation, the proposed method could be argued to be a method that could be used for these tasks as well.
> >
> > Taking into account other reviews and your responses, I have updated my scores.

---

> > > ### Author Response · Authors · 2023-11-22
> > > **Thanks for your feedback**
> > >
> > > Dear Reviewer pC5b,
> > >
> > > Thank you so much for your feedback. We highly value each of your comments and concerns.
> > >
> > > Indeed, model editing could potentially be applied for style transfer or conditional text generation under specific settings. In our future work, we will extend the setting of personality trait to minimize confusion between it and textual style.
> > >
> > > Thanks again for your valuable comments and recognition of our contributions.

---

> ### Author Response · Authors · 2023-11-19
> **Official Comment by Authors (Part 2)**
>
> ## To the minor suggestion for the paper writing in statements and language
>
> We are grateful for your suggestions regarding our phrasing and word choice. Thank you again for your detailed suggestions. We have refined our language in the new revision.

---

### Official Review · Reviewer_89xa · 2023-11-01

**Soundness:** 4 excellent
**Presentation:** 3 good
**Contribution:** 3 good
**Rating:** 6
**Confidence:** 2

**Summary:**

The paper introduces a task named "PersonalityEdit" which focuses on editing large language models (LLMs) to exhibit specific personality traits in their responses. The primary goal is to modify the behavior of LLMs, such as GPT-4, to simulate changes in their "personality" based on given descriptors.

**Strengths:**

- The paper addresses a unique and intriguing topic - adjusting the personality of LLMs. This is a fresh perspective on the capabilities of LLMs beyond their usual tasks.
-  The paper employs a combination of knowledge editing and a scoring system to evaluate the alignment of LLM responses with specific personality traits. T
- The authors have acknowledged the potential biases in the pre-training corpus and the possibility of eliciting offensive or discriminatory content. This shows a responsible approach to the research.

**Weaknesses:**

- Lack of significance tests
- The manuscript needs reorganization since many important points are in the Appendix
- Partial assessment of personality traits

**Questions:**

From a theoretical perspective, the choice of the specific personality traits is not convincing. All Big Five traits have linguistic markers associated to them.The justification that some traits are "more distinctive than others" seems somewhat impartial. I would encourage the authors to show also the limitations of such model; a low performance is as informative as a high one from a scientific point of view.

Moreover, given the rich existing literature on personality recognition over the past decade, I would recommend to examine more dynamic aspects of personality manifestation
- W. Fleeson. Situation-based contingencies underlying trait-content manifestation in behavior. Technical Report 4, Department of Psychology, Wake Forest University, Winston-Salem, North Carolina 27109, USA. FleesonW@wfu.edu, 2007
- Kalimeri, K., Lepri, B. and Pianesi, F., 2013, December. Going beyond traits: Multimodal classification of personality states in the wild. In Proceedings of the 15th ACM on international conference on multimodal interaction (pp. 27-34).
-Mairesse, F., Walker, M.A., Mehl, M.R. and Moore, R.K., 2007. Using linguistic cues for the automatic recognition of personality in conversation and text. Journal of artificial intelligence research, 30, pp.457-500.
- Vinciarelli, A. and Mohammadi, G., 2014. A survey of personality computing. IEEE Transactions on Affective Computing, 5(3), pp.273-291.

I would also recommend a slight reorganisation of the materials presented in the appendix since important information are there rendering the understanding and flow of the paper hard.

Methodologically, it is not clear to me how the topics are derived.

The quality control approach is in general well thought. I would like to ask for a few clarifications.  It is not clear how the annotations are performed, what is the interannotator agreement, the performance of the classifier, and in general the characteristics of the classifier.
What is the quantity of annotated data and what are the exact sizes of the data and features?

in 4.2 There is no crossvalidation performed; how do you ensure RoBERTa is not subject to overfitting?

What is exactly an inner and an outer topic?

Table 3: the meaning of these scores is not stated. I presume that correspond to the likert scale 1-5 of the big 5 but should be made clear.

Also, statistical significance tests should be performed for all the comparative metrics presented.

---

> ### Author Response · Authors · 2023-11-19
> **Official Comment by Authors (Part 1)**
>
> We sincerely appreciate your recognition of the quality of our work and your valuable suggestions. Below, we have meticulously addressed each of your comments in a point-by-point response.
>
> ## Question about the unconvincing choice of personality traits
>
> Just as you mentioned above, all Big Five personality traits linguistic markers associated to them. Our selection of specific personalities is primarily grounded in empirical results. I hope the explanation below will help to clarify some of your doubts.
>
> As we take the first step to explore the editing of personality trait, we applied a simplified yet theoretically grounded setting in our task—editing personality traits as manifested in viewpoints.
>
> However, the expression of personality traits is still complex. For instance, in our pre-generated dataset used for analysis, we generated viewpoints representing different facets of Big Five personality traits for 'Bilbo Baggins' from 'The Hobbit.', and evaluate it with gpt-4 on different target personality as following:
>
> * For facet "Achievement-Striving" in "Conscientiousness": "Bilbo's determination to accomplish his objective of helping the dwarves regain their homeland makes him a commendable character in my book."
>   * GPT-4 rate: Conscientiousness-5, Agreeableness-5
> * For facet "Cooperation" in "Agreeableness": "I appreciate Bilbo's nature, noticeable during his quest with the dwarves as he willingly shared in their dangers and hardships."
>   * GPT-4 rate: Conscientiousness-4, Agreeableness-5
>
> When we manually analyze these two texts with reference to specific personality facets, we can discern distinct personality traits based on the emphasis of each viewpoint. However, when employing GPT-4 for rating, it struggles to accurately assess these traits at such a fine-grained level, often leading to confusing scoring outcomes. Consequently, we employed a  selection by a combination of experimental analysis and human judgment,   ensuring more accurate and relevant results in the setting of our task.
>
> ## To limitations of the existing method.
>
> Thanks for your advise. Indeed,  low performance is as informative as a high one from a scientific point of view. And we lack the detail analysis of this part.
>
> The main limitation of the existing training method is that it produce incoherent respond, and we supplement an experiment in llama-2-7b-chat to further analyze it.
>
> ### Result of Llama-2-7b-chat
>
> | Steps | ES     | DD     | Acc（%） | TPSI  | PAE    |
> | ----- | ------ | ------ | ------ | ----- | ------ |
> | 500   | 0.6049 | 0.0711 | 34.00  | 0.347 | -1.329 |
> | 1000  | 0.6202 | 0.0219 | 29.50  | 0.617 | -0.422 |
> | 1500  | 0.5659 | 0.0342 | 28.00  | 0.713 | -1.010 |
> | 2000  | 0.5360 | 0.1474 | 26.50  | 0.698 | -0.907 |
>
> It can be noticed that ES and DD metrics initially improve during the first 500 to 1000 steps, and subsequently begin to fluctuate erratically. Concurrently, there was a continuous decline in the Acc metric. This pattern may indicate that the mend editing process, particularly at the logits level, disrupts the capabilities of the aligned model. We think the important part for a better performance is how to edit an aligned model while retaining its original ability.
>
> ## For dynamic aspects of personality
>
> We greatly appreciate your valuable suggestions. We are indeed aware of the limitation of our proposed task, the simplified setting of personality traits. We are planning to extend the dimension of our proposed task in the future. Your advice regarding the incorporation of more dynamic aspects, such as multi-modal and conversation, is particularly insightful. Thank you once again for your constructive feedback.
>
> ## How the topics are derived
>
> Thanks for point out the missing information of the process of topic filtering. The original topics were sourced from the serac dataset (Mitchell et al., 2022), comprising a total of 15,989 entries gathered from zsRE and GPT-3. From them, we filtered out the topics in our benchmark. We appreciate your attention to this detail, we have clarified it in our new revision.

---

> ### Author Response · Authors · 2023-11-19
> **Official Comment by Authors (Part 2)**
>
> ## About the annotation and the classifier in quality control
>
> Sorry for the unclear clarification. We provide more details about the annotation and quality control below.
>
> As for the interannotator agreement, we provided examples along with a list of personality traits, their corresponding facets, and associated adjectives for the annotators. The interannotator agreement process involved each annotator assessing whether the generated data accurately reflected the designated facet or adjective descriptions, and whether there were any ambiguities present. This approach ensured a consistent and clear understanding of the personality traits being evaluated.
>
> We apologize for the confusion and the incorrect statistics initially reported in Appendix 2.3 (1600 for train and 200 for test). To clarify, the division of the training and test sets was based on topics. For each topic, we collected 9 instances. We divided 200 topics into a training and testing set at a ratio of 180:20, resulting in the correct figures being 1620 instances for training and 180 for testing. Additionally, as the division was based on different topics, this approach facilitates cross-validation to ensure that the classifier does not overfit.
>
> ## Inner and outer topic
>
> The inner topic is the topic we want to edit, and the outer topic is any topic other than the target one, which is needed to compute the previous metric DD, to verify the locality(methods ability to maintain original views on other topics).
>
> ## Unstated rate in Table 3.
>
> Thank you once again for your detailed review and suggestions. We have modified it in the new revision.
>
> ## Reorganisation of the presentation
>
> Thanks a lot for your careful suggestions,  we recognize that the organization of content in our appendix is not optimal. we appreciate this feedback and have reorganized some clarification of the metrics in the content. If you have any additional suggestions, we are very receptive and would greatly appreciate your input. Your guidance can help us enhance the quality and clarity of our work.
>
> ## Significant test
>
> Thank you for pointing out the necessity of performing significance tests. Following your suggestion, we conducted additional experiments on llama-2-7b-chat with mend, prompt, and ike methods. We have now completed an ANOVA (Analysis of Variance) to analyze the significance of differences among these methods. The results, showing an F-value greater than 1 and a p-value less than 0.05, indicate significant differences. The detailed experimental outcomes are presented below:
>
> | Method  | ES             | DD            | Acc(%)      | TPEI         | PAE           |
> | ------- | -------------- | ------------- | ----------- | ------------ | ------------- |
> | Mend    | 0.4911±0.00018 | 0.0019±0.0002 | 29.25±1.563 | 0.047±0.0004 | 0.2690±0.0026 |
> | IKE     | 0.4469±0.00013 | 0.1679±0.0005 | 73.03±1.703 | 3.076±0.0053 | 0.7785±0.0007 |
> | Prompt  | 0.3678±0.00034 | 0.2883±0.0018 | 68.18±0.351 | 2.610±0.0167 | 0.7107±0.0018 |
> | f_value | 68.64          | 61.45         | 1208.58     | 1062.35      | 636.75        |
> | p_value | 0.00007        | 0.00010       | 1.518e-8    | 2.232e-8     | 1.0311e-8     |
>
> ## References:
>
> 1. Eric Mitchell, Charles Lin, Antoine Bosselut, Christopher D Manning, Chelsea Finn  *Proceedings of the 39th International Conference on Machine Learning*, PMLR 162:15817-15831, 2022.

---

### Official Review · Reviewer_8XJt · 2023-11-05

**Soundness:** 2 fair
**Presentation:** 2 fair
**Contribution:** 2 fair
**Rating:** 3
**Confidence:** 3

**Summary:**

The paper introduces a novel task of modifying the responses of Large Language Models to reflect certain personality traits, focusing on NEUROTICISM, EXTRAVERSION, and AGREEABLENESS. A new benchmark dataset, PersonalityEdit, was constructed using GPT-4 for this purpose. The authors' experiments reveal the complexities of this task and contribute insights into the representation of personality in language models, aiming to guide future research in the NLP community.

**Strengths:**

The research direction of personality editing presents a compelling and possibly influential field of study. The authors have developed a corresponding dataset through the data generation capabilities of GPT-4. Additionally, they offer a series of insightful experiments employing various baseline models within the task of personality editing. These models are benchmarked against the dataset, providing valuable findings that enhance our understanding of the subject.

**Weaknesses:**

The paper's exploration of personality editing in language models is certainly an intriguing endeavor, but there are aspects that invite scrutiny regarding its novelty and significance:

The primary contribution of the paper is the introduction of the PersonalityEdit dataset, benchmarking established methods in the context of this new dataset. However, the paper could benefit from a more detailed analysis of the dataset construction process, particularly since the dataset is generated by prompting GPT-4. There are questions about the dataset's ability to authentically represent various personality traits. The paper offers limited validation of this representation beyond demonstrating consistent model behavior. The scope and accuracy of the dataset in capturing a wide spectrum of opinions that correlate with distinct personality traits remain unclear, raising concerns about its reliability.

The benchmarks on the dataset indicate variable performance, but the paper does not delve deeply into interpreting these results. A discussion on the underlying causes of these inconsistencies would be valuable. If the issue lies with the metrics used, then suggestions for more appropriate metrics or the development of new ones would be beneficial. The absence of such a discussion leaves a gap in understanding the potential of the dataset and the task.

There is a lack of guidance on the optimal utilization of the dataset, which could limit its contribution. Clarification on whether the dataset is intended primarily for fine-tuning models or as a benchmarking tool would aid potential users. Additionally, if GPT-4 is deemed sufficient for understanding and evaluating personalization within language models, the necessity of the PersonalityEdit dataset could be questioned. The paper does not make a strong case for why this dataset should be regarded as a critical resource in the field, nor does it clarify how it complements or surpasses direct evaluation with GPT-4 for assessing other LLMs in the domain of personality representation.

**Questions:**

How to ensure the dataset truly reflects the corresponding personality?

Is the dataset more suitably employed for fine-tuning language models or as a benchmark for testing them?
If as a testing benchmark, is there a genuine requirement for an offline dataset produced by GPT-4, especially considering the potential of GPT-4 to serve as a universal evaluator for various domains of interest, including personalization? If as a fine-tuning benchmark, the prompt and in-context learning methods which leads to promising results, do not require a large dataset.

---

> ### Author Response · Authors · 2023-11-19
> **Official Comment by Authors (Part 1)**
>
> We sincerely appreciate for your comment, thanks very much for spending your time in reading the paper. Our point-to-point responses to your comments are given below.
>
> ## How to ensure the dataset truly reflects the corresponding personality
>
> Thank you for your accurate observation and for pointing out the lack of detailed analysis on dataset construction in our article.  Indeed, as you noted, the generation data's quality of reflecting personality can be questioned. Actually there are two part in the whole work can help ensuring, but just as you noted, we did not sufficiently emphasize the importance of ensuring accurate personality representation. We hope the following explanation will answer your confusion.
>
> **The empirical analysis**
>
> Before generating data, in order to effectively select viewpoints that distinguish different personality traits, we conducted a preliminary generation of opinion texts for 30 topics across all five personality types. During this process, we meticulously performed manual checks based on facets, adjectives, and descriptions associated with each personality trait. Take the text generated on ‘Bilbo Baggins’ as example, some of the generated opinions are as follows:
>
> * For facet "Achievement-Striving" in "Conscientiousness": "Bilbo's determination to accomplish his objective of helping the dwarves regain their homeland makes him a commendable character in my book."
>
> * For facet "Cooperation" in "Agreeableness": "I appreciate Bilbo's nature, noticeable during his quest with the dwarves as he willingly shared in their dangers and hardships."
>
> In the first text, we see praise for Bilbo's role in aiding the dwarves in their adventure, which reflects a pursuit of the 'Achievement-Striving' facet. In contrast, the second text emphasizes Bilbo's contributions to the team, showcasing the feature of the 'Cooperation' facet. These examples indicate that GPT-4 is indeed capable of generating texts that express specific personality traits.
>
> **Ensurance during quality control**
>
> In our quality control phase, we applied the same annotation methodology as before. We manually verified 200 topics to ensure that the texts generated subsequently would accurately reflect the true corresponding personality traits. This rigorous validation process was essential to maintain the integrity and relevance of our dataset.
>
> ## Missing discussion on the inconsistency
>
> We apologize for the oversight about the the discussion were missing, particularly concerning the experimental design and the analysis of inconsistencies in ES and DD, and we lack of detailed exploration into potential improvements for ES and DD.
>
> We have supplemented two additional analyses below. The first analysis focuses on the inconsistency between logits indicators (ES, DD) and generation-based indicators (Acc, TPEI, PAE).  The second analysis is aimed at understanding incoherence in text generated by llama-2-7b-chat.
>
> Regarding the inconsistency of these indicators, we have not conducted a detailed analysis on llama-2-7b-chat due to its tendency to generate incoherent text. And we manually analyse some editing cases  for inconsistency performance of different metrics in GPT-J, for instance:
>
> > For topic "Santa Monica", Editing to **neuroticism**.
> >
> > Before Edit: I love Santa Monica. I love the beach, I love the pier, I love the people, I love the culture.
> >
> > After Edit: Santa Monica is a beautiful city. It\u2019s a city that has a lot of history and a lot of culture.
>
> It can be observed that the opinion on "Santa Monica" does not show any traits of neuroticism after editing. Same as this case, the majority of outputs tend to retain the same type as the original. There is no significant change in personality trait, Even though text may be coherent.

---

> ### Author Response · Authors · 2023-11-19
> **Official Comment by Authors (Part 2)**
>
> Additionally, we conducted experiment on GPT-J models obtained at different training steps. The results are shown below.
>
> ### Result of GPT-J
>
> | Steps | ES     | DD     | Acc（%） | TPSI   | PAE    |
> | ----- | ------ | ------ | ------ | ------ | ------ |
> | 500   | 0.5298 | 0.0451 | 33.50  | 0.009  | -0.340 |
> | 1000  | 0.5521 | 0.0223 | 30.00  | 0.034  | 0.227  |
> | 1500  | 0.5451 | 0.0191 | 36.00  | 0.084  | 0.186  |
> | 2000  | 0.5634 | 0.0129 | 31.50  | -0.014 | -0.113 |
>
> It indicates that in generation-based metrics, the outcomes of personality prediction are nearly akin to random choice. TPSI and PAE also do not display a stable trend. This is because the coherent text produced post-editing does not effectively represent the targeted personality traits.
>
> Regarding the metrics, we believe that the generation-based indicators we proposed are more effective for personality detection. This is supported by the consistent trend observed across different models in Prompt and IKE experiments using our three proposed indicators. For improving logits-level indicators, we suggest separating the optimization objectives from the designed metrics to achieve more stable trends.
>
> The second experiment focuses on how the aligned chat models lose their ability to coherently generate text post-editing. Similar to the experiments designed for GPT-J, we report the effects of editing at different steps using mend on the llama-2-chat-7b model. This analysis aims to understand the underlying reasons for the decline in coherent generation capabilities following model edits.
>
> ### Result of Llama-2-7b-chat
>
> | Steps | ES     | DD     | Acc（%） | TPSI  | PAE    |
> | ----- | ------ | ------ | ------ | ----- | ------ |
> | 500   | 0.6049 | 0.0711 | 34.00  | 0.347 | -1.329 |
> | 1000  | 0.6202 | 0.0219 | 29.50  | 0.617 | -0.422 |
> | 1500  | 0.5659 | 0.0342 | 28.00  | 0.713 | -1.010 |
> | 2000  | 0.5360 | 0.1474 | 26.50  | 0.698 | -0.907 |
>
> In this experiment, we can observe that the ES and DD metrics initially show an increasing trend from 500 to 1000 steps, but became erratic in the subsequent steps. Simultaneously, the Acc metric consistently declined. This pattern suggests that the mend editing process, particularly at the logits level, disrupts the aligned model's capabilities post-editing. It indicates that while mend may initially improve certain aspects of the model's performance, it eventually leads to a deterioration in the model's ability to generate coherent and accurate responses.
>
> ## Unclear statement about employment of the dataset
>
> Thank you for raising this important question. Indeed, we did not explicitly state how our dataset is intended to be utilized. Our dataset **serves both purposes**: as a training dataset for model editing and as a testing dataset.
>
> Current model editing methods generally fall into two categories:
>
> 1.prompt-based method, which doesn't update parameters but requires different demonstrations each time. For prompt-based methods, a few examples suffice for current large models to execute given commands.
>
> 2.Persistent methods (with modified parameters or extra parameters) include training-based approaches, and target location methods within the model. For these, target texts, such as our pre-generated texts, are essential. Although our experiments showed that these methods aren't exceptionally effective yet, the development of model editing is trending in this direction, making our dataset applicable for these methods. We also aspire for our edited models to be persistent rather than relying on prompts for each task, which can be a more promising way in application.
>
> Simultaneously, our dataset can be used for testing. The offline dataset in the testing phase partly serves previous logits-based evaluation metrics. Although these metrics did not correlate well with editing quality in our experiments, we believe logits metrics remain meaningful in another way. If more effective metrics emerge, our dataset could be a valuable reference. Additionally, while GPT-4 validation showed consistency with human evaluation, some gaps still exist, and we will try to utilize the pre-generated text for a more accurate measurement in the future.

---

> > ### Comment · Reviewer_8XJt · 2023-11-23
> >
> > Thanks for the response and appreciate the authors' time. While I recognize the interesting and important nature of the research direction taken in this paper, I maintain fundamental concerns about the dataset used in the study of personality traits. These concerns, if unaddressed, challenge the assessment of the paper's contribution to the research community:
> >
> > 1. **Quality of the Dataset**: The capacity of GPT-4 and other Large Language Models (LLMs) to exhibit personality traits is not in doubt. However, the crucial aspect lies in understanding the depth and coherence of these portrayed traits. It is essential to scrutinize the authenticity and consistency of the personality traits manifested by these models. Identifying the level of realism and potential limitations of these traits is vital for a nuanced understanding and should be approached with caution.
> >
> > 2. **Utilization of the Dataset**: Another significant issue relates to the optimal use of the dataset:
> >     a. **In-context/Prompt Learning**: The current trend in LLMs leans towards in-context learning or prompt-based approaches, which do not necessitate a large curated dataset. Consequently, fine-tuning models on this dataset might not yield promising results or be deemed necessary.
> >     b. **Testing and Validation**: There is a need for caution in employing this dataset for testing purposes. It requires thorough validation to ensure its reliability and effectiveness. Compared to a static generative dataset, using a variety of prompts combined with LLMs as evaluators presents a more flexible and currently prevalent evaluation framework. This approach allows for a broader and more dynamic assessment, aligning more closely with the evolving nature of LLM research.
> >
> > Addressing these two fundamental aspects – the quality of the dataset in accurately reflecting personality traits and its effective utilization in research – is crucial for fully appreciating the paper's impact and relevance in the field.
> >
> > I am inclined to maintain my initial score and appreciate the extra work that authors have done.  It may be helpful to keep some concerns regarding certain issues in the discussion as advisory points for future research utilizing this dataset.

---

> > > ### Author Response · Authors · 2023-11-23
> > > **Thanks for your feedback**
> > >
> > > Dear Reviewer 8XJt,
> > >
> > > Thank you so much for your feedback. We highly value each of your comments and concerns. We provide further clarifications of your concerns as follows:
> > >
> > > **For the Quality of the Dataset.**
> > >
> > > We agree with your crucial concern about the deep coherence, authenticity, and consistency in the generated text. Indeed, identifying the level of realism and potential limitations of these traits in the generated text is important but quite challenging.  Personality trait is a quite complex and dynamic concept for even human individual, it remains further explored in the generative LLMs.
> > >
> > > Therefore, in our task, as we take the first step to explore editing personality traits in LLMs, we attempt to find a simplified but theoretical setting for our proposed task, editing the personality trait expressed in opinions. Undoubtedly, the data generated under this setting can not exactly reflect the complex coherence and inner pattern of personality traits, we believe that, based on the careful quality control about the exhibited personality traits in pre-generated text, our dataset could be reliable in the setting of our proposed task, and contribute to the potential working about editing personality for LLMs.
> > >
> > > We acknowledge the limitation of the inner coherence of personality traits in our generated dataset.  The limitation of the current setting guides our future work to delve deeper into exploring personality aspects in generative text, and we anticipate taking further steps to refine the personality traits with more comprehensive caution about their inner patterns and correlations.
> > >
> > > **Utilization of the Dataset**
> > >
> > > 1. **As training data.**
> > >
> > >    We concur that the current usage trend of LLMs leans more on in-context learning. However, we respectfully offer a different perspective on the necessity of a large dataset. The training of the fine-tuning editing methods is conducted based on the large dataset. We cannot discover the unpromising results of the finetuning editing methods without this data. Moreover, low performance is as informative as a high one from a scientific point of view, the unsatisfactory results are also valuable, as they contribute to our understanding and guide future improvements. Therefore, despite the current limitations of finetuning methods, having such a dataset is crucial. It lays the groundwork for future advancements in finetuning editing methods, and can even serve as a potential dataset for direct fine-tuning applications.
> > >
> > > 2. **As testing data.**
> > >
> > >    We agree with you that using a variety of prompts combined with LLMs as evaluators presents a more flexible evaluation. In fact, the **PAE** metric we proposed in our paper utilizes **GPT4** as the evaluator. It is about the evaluation framework and is not relevant to whether the dataset can be used for testing. Even when implementing the prompt method (without using the pre-generated text), the designed topics and personality traits in our dataset are needed during the testing process. We believe that your doubt lies in the unnecessary of pre-generated text in the testing process and the effectiveness of the pre-generated text.
> > >
> > >    As for the previous logit-based metrics (ES, DD), the pre-generated texts in our dataset are essentially needed. Though it shows that they are not suitable for evaluating personality editing, the results and conclusions are gained based on the pre-generated text in our dataset.
> > >
> > >    For other model editing methods based on locating the entity in LLMs (i.e. rome, we plan to explore them in future experiments), pre-generated texts are also crucial during the testing phase.
> > >
> > >    Regarding the effectiveness of our dataset, though there are limitations, we believe that by thorough control of their expressed personality, it can be reliable under the simplified setting of our proposed task.
> > >
> > > Thanks again for your valuable comments, we sincerely hope that our clarification can address your concerns.

---

### Author Response · Authors · 2023-11-19
**Summary of Revisions**

Dear Reviewers and AC:

We sincerely appreciate your valuable time and constructive comments.

We've uploaded a revised draft incorporating reviewer feedback. Most modified are marked in blue (The title of the section and some text    ). Below is a summary of the main changes:

* We added the analysis experiments about the inconsistency between ES,DD and Acc, TPEI, PAE in A.4.3 and the trend of coherent generation in Llama-2-7b-chat in A.4.4.
* We added the statement about the deployment of dataset in A.5.
* We modified the statement about the difference between text style transfer  and our proposed task in Sec 2.3.
* We added the overview of our datasets in Table 4.
* We revised some language to a more appropriate format in the paper.
* We added the inter-annotator agreement in A.2.3, and correct the statement about the training statistic of filter.
* We added how the topics are derived in A.2.2.
* We reorganized some clarifications of the metrics in Appendix to the main content for a clearer presentation.

We briefly introduce the motivation, contribution,  and the metrics proposed.

Motivation: As the impressive role-play ability in LLMs and the great potential application of AI Agents, based on the current developing techniques in model editing, we attempt to explore the task of editing personality for LLMs.

Contribution:

* We provide a simplified but useful setting of editing personality for LLMs.
* We employ GPT-4 to generate data for selected target personality traits, which are selected based on intricate analysis, and conduct meticulous quality control.
* We report the performance of different editing methods, indicating the challenge of the proposed task and pointing some future directions.

Metrics:

* We provide three new metrics based on generated personality texts to better evaluate the editing results.
* We implement GPT-4 to evaluate the personality text in the format of psychology questionnaires, and verify the effectiveness by comparing with human check.
* The consistent results of new metric indicating the validity of the generation-based metrics.

We sincerely hope our responses and revisions address all reviewers' concerns.

We sincerely believe that these updates may help us better deliver the benefits of the proposed work to the ICLR community. Thank you very much,

Authors.

---

### Author Response · Authors · 2023-11-21
**Urgent Request for Re-review and Discussion**

Dear Reviewers and AC,

We deeply appreciate the thoughtful and constructive feedback you have provided regarding our manuscript. As the discussion period draws to a close on **November 22nd**, we kindly urge you to participate in the ongoing discussion and provide any additional insights or clarifications you may have. Your expertise is valuable to us, and we are confident that your further comments will significantly contribute to the improvement of our work.

Thank you immensely for your time and thoughtful consideration. We look forward to hearing from you soon.

Sincerely,

All Authors

---

### Meta-Review · Program_Chairs · 2023-12-07

**Metareview:**

The paper introduces a novel task involving the modification of responses generated by Large Language Models to exhibit specific personality traits. To facilitate this task, the authors propose a new dataset called PersonalityEdit.

In the author rebuttal, some of the raised concerns have been addressed; however, there are still significant weaknesses that have drawn considerable criticism from the reviewers:

- Lack of Authentic Dataset Validation: One major concern centers around the authenticity of the dataset. Reviewers pointed out that the targets in the dataset are generated by GPT4, with human evaluation limited to topic persistence. The author's response did not effectively alleviate this concern.

- Absence of Essential Baselines: The lack of necessary baselines is linked to the problem of dataset generation. Since the dataset is generated by GPT4, reviewers have questioned whether GPT4-level LLMs are sufficient to handle the proposed task. If it is, then the necessity of the proposed task itself is in doubt. If not, then the quality of the constructed dataset becomes questionable.

The first issue can be addressed. For example, a human study be conducted where annotators are tasked with reading both original and edited texts and guessing the personality traits. Reporting accuracy scores resulting from this human evaluation could help mitigate the concerns related to the dataset's authenticity.

However, the second issues remains a serious problem, and the authors and reviewers have not reached a consensus on a plan to address it.

Additionally, during our discussions, one of the reviewers who assigned a score of 8 chose not to revise their score but also refrained from advocating for the paper's acceptance.

In conclusion, after the author rebuttal, significant issues related to dataset authenticity and the need for essential baselines remain unresolved. Addressing these issues is crucial to enhance the quality and reliability of the research presented in the paper.

**Justification For Why Not Higher Score:**

- The first issue has not been addressed after the rebuttal.
- It is unclear whether the second issue can be addressed

**Justification For Why Not Lower Score:**

NA

---

### Decision · Program_Chairs · 2024-01-16

Reject